# mTORC1 in AGRP neurons integrates exteroceptive and interoceptive food-related cues in the modulation of adaptive energy expenditure in mice

Luke K Burke[1,2], Tamana Darwish[1,2], Althea R Cavanaugh[3†], Sam Virtue[1,2], Emma Roth[1,2], Joanna Morro[1,2], Shun-Mei Liu[3], Jing Xia[4], Jeffrey W Dalley[4,5], Keith Burling[1,2], Streamson Chua[3], Toni Vidal-Puig[1,2], Gary J Schwartz[3], Clémence Blouet[1,2]*

[1]MRC Metabolic Diseases Unit, Metabolic Research Laboratories, University of Cambridge, Cambridge, United Kingdom; [2]WT-MRC Institute of Metabolic Science, University of Cambridge, Cambridge, United Kingdom; [3]Departments of Medicine and Neuroscience, The Albert Einstein College of Medicine, New York, United States; [4]Department of Psychology, Behavioural and Clinical Neuroscience Institute, University of Cambridge, Cambridge, United Kingdom; [5]Department of Psychiatry, Behavioural and Clinical Neuroscience Institute, University of Cambridge, Cambridge, United Kingdom

*For correspondence: csb69@medschl.cam.ac.uk

Present address: †Solomon H. Snyder Department of Neuroscience, Johns Hopkins University, Baltimore, United States

Competing interests: The authors declare that no competing interests exist.

**Abstract** Energy dissipation through interscapular brown adipose tissue (iBAT) thermogenesis is an important contributor to adaptive energy expenditure. However, it remains unresolved how acute and chronic changes in energy availability are detected by the brain to adjust iBAT activity and maintain energy homeostasis. Here, we provide evidence that AGRP inhibitory tone to iBAT represents an energy-sparing circuit that integrates environmental food cues and internal signals of energy availability. We establish a role for the nutrient-sensing mTORC1 signaling pathway within AGRP neurons in the detection of environmental food cues and internal signals of energy availability, and in the bi-directional control of iBAT thermogenesis during nutrient deficiency and excess. Collectively, our findings provide insights into how mTORC1 signaling within AGRP neurons surveys energy availability to engage iBAT thermogenesis, and identify AGRP neurons as a neuronal substrate for the coordination of energy intake and adaptive expenditure under varying physiological and environmental contexts.

## Introduction

In mammals, the maintenance of energy balance relies on a tight coordination of energy intake and energy expenditure. Internal energy surfeit, typically achieved via maintenance on energy dense food in a laboratory setting, promotes recruitment of interscapular brown adipose tissue (iBAT) thermogenesis, increasing energy expenditure and therefore limiting weight gain (*Bachman et al., 2002*; *Feldmann et al., 2009*). Conversely, energy expenditure is robustly decreased in energy-restricted rodents and humans (*Ravussin et al., 2012*; *Leibel et al., 1995*). This protective mechanism mitigates weight loss when environmental energy sources are limited and may contribute to the failure of long-term maintenance of weight loss through dieting or pharmacological interventions. The development of successful therapies to treat obesity therefore requires a better

**eLife digest** Losing weight through dieting can be difficult. Weight loss strategies often prove ineffective because the body works like a thermostat and couples what we eat to the number of calories we burn. When we eat less, our bodies compensate and burn fewer calories, which makes losing weight harder. The brain is the master regulator of this caloric thermostat, but it is not clear how it adjusts our energy expenditure to account for how much we have eaten.

A structure deep within the brain called the hypothalamus, which helps regulate appetite, is thought to be involved in the caloric thermostat. Activating a group of neurons within the hypothalamus called the agouti-related neuropeptide (AGRP) neurons causes animals to consume large quantities of food. By contrast, inhibiting AGRP neurons causes animals to stop eating almost entirely.

Burke et al. studied AGRP neurons in mice. The experiments show that artificially activating the neurons in mice that don't have access to food increases the animals' activity levels but reduces the rate at which they burn calories, which helps the mice to maintain their existing weight. Allowing the mice to eat, or even just to see and smell food, switches off this effect and returns energy expenditure to normal. Finally, exposing mice to a high-fat diet for several days inhibits their AGRP neurons, and causes the animals to burn calories at a faster rate. By using up excess calories, this change also helps the animals maintain their existing body weight.

The findings of Burke et al. show that AGRP neurons are a key component of the caloric thermostat. By adjusting the rate at which the body burns calories, AGRP neurons can compensate for any changes in food intake and so limit changes in body weight. This work opens up the possibility of developing therapies that disconnect energy expenditure from energy intake to help maintain long-term weight loss.

characterization of the mechanisms underlying the regulation of adaptive energy expenditure in various contexts of energy availability.

Sympathetic nervous outflow to iBAT modulates adaptive energy expenditure by promoting iBAT thermogenesis (*Bartness et al., 2010*) and is modulated by internal energy availability (*Brito et al., 2008*). Melanocortinergic circuits, one of the best characterized central energy-sensing network regulating energy balance, have been implicated in the control of sympathetic outflow to iBAT (*Bartness et al., 2010*; *Small et al., 2003*), but the exact neuronal mechanisms and energy-sensing pathways that modulate sympathetic tone to iBAT during caloric restriction or nutrient excess have not been identified.

Orexigenic neurons expressing agouti-related peptide (AGRP) and neuropeptide Y (NPY) form a discrete population of about 10,000 cells residing in the arcuate nucleus of the mouse hypothalamus (ARH). These neurons respond to bidirectional changes in energy availability, are activated by fasting, low glucose levels and ghrelin, and are inhibited by leptin, insulin, refeeding and high glucose levels (*Chen et al., 2015*; *Könner et al., 2007*). Their essential role in the bidirectional control of energy intake can be appreciated by the aphagia observed in mice with specific ablation of this neuronal population during adulthood (*Luquet et al., 2005*; *Gropp et al., 2005*), and the voracious feeding response upon opto- or chemogenetic activation of AGRP neurons in sated mice (*Aponte et al., 2011*; *Krashes et al., 2011*). Initial evidence indicates that AGRP neurons also regulate energy expenditure (*Krashes et al., 2011*), but the contribution of iBAT thermogenesis to this effect, the physiological contexts in which this axis is engaged, and the mechanisms through which AGRP neurons monitor energy availability have not been identified.

Here, we used chemogenetics to rapidly and selectively activate AGRP neurons, and demonstrate that AGRP neurons can engage an energy-sparing circuit to iBAT that represents the integration of metabolic interoception and environmental food-related cues. We show that mTORC1 signaling in AGRP neurons surveys sensory food cues and internal signals of energy availability. Using inducible lentivectors to manipulate the nutrient-sensing mTORC1 signaling pathway specifically in AGRP neurons of adult normally developed mice, we show that mTORC1 signaling within AGRP neurons acts as a bidirectional node of control that modulates the AGRP-iBAT axis. Collectively, our data describe

a novel neuronal energy sensing mechanism that modulates sympathetic tone to iBAT and identify AGRP neurons as a neuronal substrate for the coordination of energy intake and adaptive expenditure under various physiological and environmental contexts.

## Results

### Activation of AGRP neurons rapidly inhibits sympathetic tone to iBAT and suppresses iBAT thermogenesis

To rapidly and reversibly activate AGRP neurons, we used the stimulatory hM3dq designer receptor exclusively activated by designer drugs (DREADD) (*Alexander et al., 2009*). This receptor couples through the Gq pathway upon exposure to the otherwise pharmacologically inert ligand clozapine-N-oxide (CNO), leading to neuronal depolarization. *Agrp-IRES-cre* mice received a bilateral stereotactic injection of an adeno-associated virus expressing hM3dq (AAV-hM3dq-mCherry), and were implanted with a telemetric probe under the interscapular adipose depot to remotely monitor iBAT temperature in behaving animals. Successful stereotactic targeting of hM3dq in AGRP neurons was confirmed by ARH mCherry staining (*Figure 1—figure supplement 1a*), as well as with a functional feeding test. Mice selected for subsequent studies ate at least 2-fold more during the first hour following the CNO injection as compared to the vehicle injection (*Figure 1—figure supplement 1b*).

Previous reports indicate that chemogenetic activation of AGRP neurons rapidly increases food intake and suppresses oxygen consumption (*Krashes et al., 2011*). To determine the direct effect of AGRP activation on adaptive energy expenditure independently of energy intake, we tested the effect of AGRP activation on iBAT temperature and energy expenditure in mice that were food deprived for 2 hr, followed by an ip injection of CNO or vehicle without access to food following the injection (Design 1). As expected, activation of AGRP neurons rapidly suppressed iBAT temperature (*Figure 1a*) and energy expenditure (*Figure 1b*) despite increasing locomotor activity (*Figure 1c*) when mice were maintained in a food-free environment following CNO administration. Local administration of ghrelin, an endogenous activator of AGRP neurons, into the mediobasal hypothalamus (MBH) of wild-type mice decreased iBAT temperature (*Figure 1—figure supplement 1c*), supporting the physiological relevance of this effect.

We investigated the mechanisms mediating the rapid fall in iBAT thermogenic activity following AGRP neuronal activation by first assessing sympathetic outflow to iBAT. iBAT norepinephrine turnover (NETO) was measured using the $\alpha$-methyl-p-tyrosine (AMPT) method. In CNO-treated mice, iBAT NETO was significantly lower than in vehicle-treated mice (*Figure 1d*), supporting the conclusion that AGRP activation rapidly suppresses sympathetic output to iBAT. In contrast, circulating thyroxine (T4) and FGF-21 levels remained unchanged during the first 45 min following CNO administration (*Figure 1—figure supplement 1d and e*), ruling out the contribution of these hormones in the rapid decrease in iBAT thermogenesis following AGRP neuronal activation. Consistent with previous reports (*Ruan et al., 2014*), we did not observe any changes in iBAT expression of thermogenic genes within an acute 1 hr time-frame following CNO administration (*Figure 1—figure supplement 1f*). These results indicate that iBAT recruitment is not involved in the rapid decrease in iBAT temperature observed as soon as 1 hr after AGRP neuronal activation.

We examined the activation status of signaling pathways downstream from $\beta$3-adrenergic receptor in iBAT 1 hr after CNO administration. We did not detect changes in the expression or activation status of PKA, CREB, perilipin, UCP-1 or in the activating phosphorylation of HSL on Ser[660] (*Figure 1—figure supplement 1g*). In contrast, inhibitory phosphorylation of HSL on Ser[565] was significantly increased in the iBAT of CNO-treated mice (*Figure 1e*). AMP-activated protein kinase (AMPK) is inhibited by $\beta$3-adrenergic signaling in iBAT and has been reported to phosphorylate HSL on Ser[565] phosphorylation (*Anthony et al., 2009*; *Pulinilkunnil et al., 2011*). This led us to hypothesize that decreased sympathetic tone to iBAT in response to AGRP activation may activate AMPK and reduce iBAT thermogenesis via an increase in HSL inhibitory phosphorylation on Ser[565]. Accordingly, we observed a robust increase in AMPK activatory phosphorylation in the iBAT of CNO-treated mice (*Figure 1e*). iBAT thermogenesis is an important contributor to thermoregulation in mice housed at ambient temperature. To determine whether activation of AGRP neurons affect thermoregulation, we simultaneously monitored iBAT and core temperature during AGRP chemogenetic activation. At baseline and in vehicle-treated animals, iBAT temperature was consistently 1°C higher than core

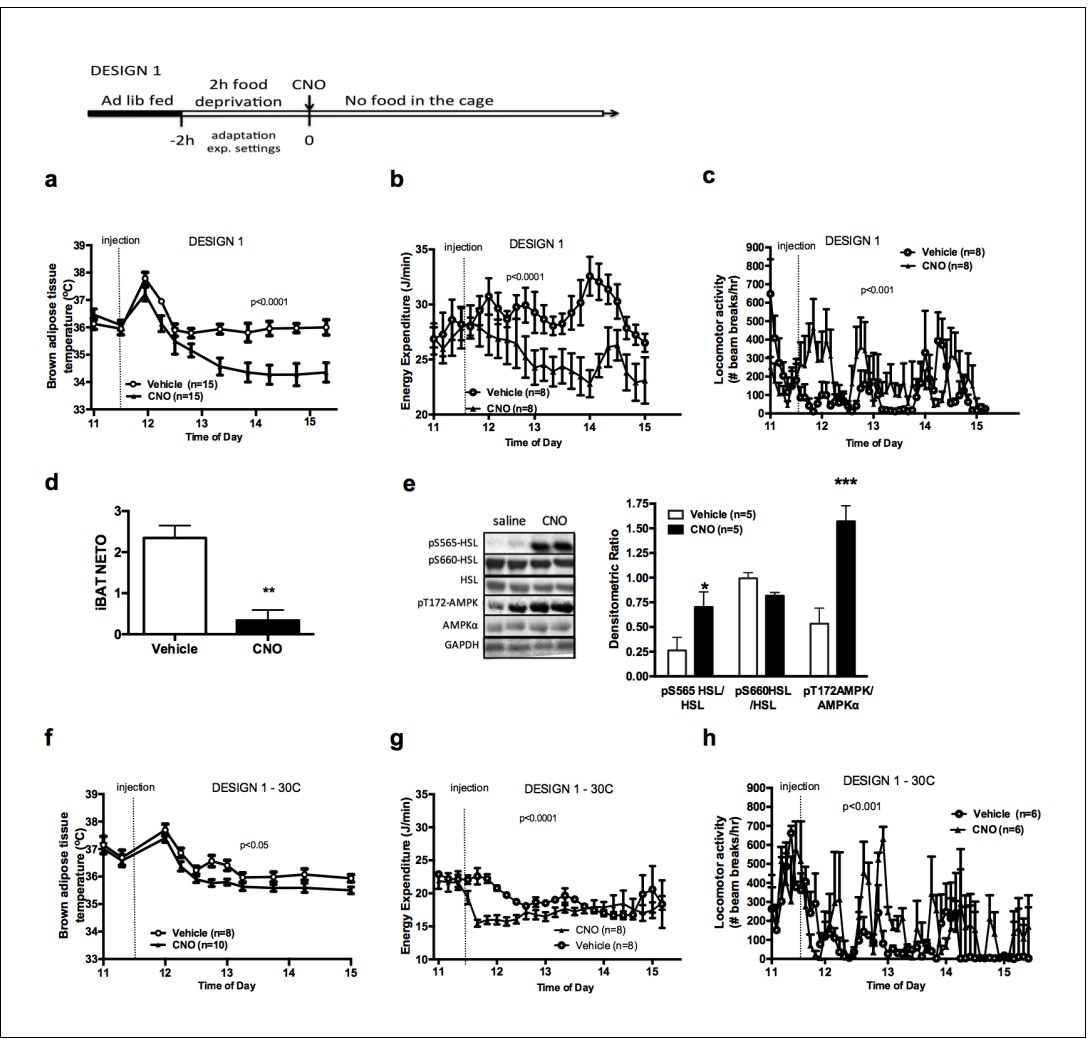

**Figure 1.** AGRP neurons control iBAT thermogenesis. iBAT temperature (**a**), energy expenditure (**b**), locomotor activity (**c**), Norepinephrine turnover (n = 8) (**d**) and phosphorylation levels of lipolytic pathways in the iBAT (**d**) following chemogenetic activation of AGRP neurons in food-deprived mice (DESIGN 1). iBAT temperature (**f**), energy expenditure (**g**) and locomotor activity (**h**) following chemogenetic activation of AGRP neurons at 30°C. Data are mean ± SEM, *p<0.05 ; ***p<0.001.

The following figure supplement is available for figure 1:

**Figure supplement 1.** AGRP neurons control iBAT thermogenesis.

temperature (*Figure 1—figure supplement 1h*). Upon CNO administration, iBAT temperature fall rapidly and reached core temperature within 15 min, indicating that suppression of iBAT thermogenesis under these conditions is driving iBAT and core temperature responses. We then tested whether the control of iBAT thermogenesis by AGRP neurons was independent of thermoregulation, and repeated these experiments in mice maintained in the thermoneutral zone (30°C), a condition under which thermoregulatory thermogenesis is abolished. Under these conditions, the effects of AGRP activation on iBAT temperature, energy expenditure and locomotor activity were attenuated, but remained statistically significant (*Figure 1f–h*). This result further directly implicates iBAT thermogenic activity in the response to AGRP activation and indicates that the AGRP-iBAT circuit can be engaged independently of thermoregulation. At 4°C, a condition under which thermoregulatory iBAT thermogenesis is strongly activated (*Brito et al., 2008*), AGRP neuronal activation reduced iBAT temperature to a similar extent as at ambient temperature (*Figure 1—figure supplement 1i*),

reinforcing the idea that the dynamic range of AGRP-iBAT circuit function is dissociable from environmental temperature and cold-induced thermoregulation. Thus, activation of AGRP neurons increases locomotor activity and produces a substantial sustained decrease in energy expenditure through a suppression of iBAT thermogenic activity.

## The regulation of iBAT thermogenesis by AGRP neurons depends on food availability and is modulated by sensory cues of environmental food availability

To better characterize the relationship between the control of energy intake and energy expenditure in response to AGRP neuronal activation, we performed experiments in mice that had access to food following CNO administration (Design 2). Under these conditions, activation of AGRP neurons increased locomotor activity and energy intake but failed to alter iBAT temperature and energy expenditure (*Figure 2a–c*), indicating that energy availability modulates the AGRP-iBAT circuit.

To test whether internal signals of energy availability were sufficient to blunt the response to CNO, mice were fasted overnight and refed before receiving an injection of CNO or vehicle, and then placed in food-free clean cages immediately after the injection (Design 3). In that context, AGRP neuronal activation suppressed iBAT temperature and energy expenditure (*Figure 2d–2f*), suggesting that internal signals of energy availability are not sufficient to blunt the AGRP-iBAT hypometabolic response.

We noticed that iBAT temperature and energy expenditure were rapidly restored in CNO-treated mice upon food reintroduction in design 1 (*Figure 2—figure supplement 1a and b*), suggesting that the presence of food rapidly resets the AGRP-iBAT circuit. To directly test the role of environmental cues of food availability in AGRP inhibitory tone to iBAT, we repeated this experiment in mice kept in clean food-free cages after CNO administration and transiently exposed to food sensory stimuli. Specifically, we introduced chow pellets in a container that allowed food to be seen and smelled but not consumed (caged-food). This paradigm was recently found to rapidly inhibit AGRP neurons in hungry mice (*Chen et al., 2015*). Presenting caged-food to CNO-treated mice rapidly increased iBAT temperature (*Figure 2g*) and energy expenditure (*Figure 2h*) to levels similar to those measured in controls, an effect that was transient and lasted for about 1 hr. Importantly, presentation of an inedible empty container to naïve animals failed to alter iBAT temperature and energy expenditure (*Figure 2—figure supplement 1c–d*), indicating that the stress response to the introduction of an object to naïve mice is not sufficient to produce the restoration in iBAT temperature seen with presentation of food sensory cues. Again, we observed that CNO administration increased locomotor activity (*Figure 2i*) despite reducing energy expenditure and iBAT temperature. These results indicate that pre-ingestive food detection is sufficient to apprehensively turn off the AGRP-iBAT circuit suppressing iBAT thermogenesis and energy expenditure. Thus, the AGRP-iBAT circuit integrates sensory and endogenous signals of energy availability in the regulation of iBAT thermogenic activity and energy expenditure to promote energy sparing when fuel availability is scarce.

## mTORC1 signaling in AGRP neurons responds to acute and chronic signals of energy abundance and environmental cues of food availability

AGRP neurons are essential to the control of energy balance and ideally positioned within and around the median eminence to detect circulating signals of energy availability. However, little is known about the intracellular energy-sensing signaling pathways coupling energy availability to neuronal activity in AGRP neurons. The energy-sensing mTORC1 pathway is an evolutionary-conserved neuronal nutritional and hormonal sensor that regulates hunger-driven eating and foraging (*Cota et al., 2006*; *Blouet and Schwartz, 2012*), two behaviors regulated by AGRP neurons (*Aponte et al., 2011*; *Krashes et al., 2011*; *Teubner et al., 2012*). We hypothesized that energy sensing through mTORC1 in AGRP neurons monitors internal energy availability to coordinate adaptive energy expenditure.

We first tested whether the activity of mTOR in NPY/AGRP neurons is modulated by a refeeding episode. This nutritional transition is associated with a rapid inhibition of AGRP neurons (*Chen et al., 2015*), an activation of iBAT thermogenic activity and an increase in energy expenditure (*Figure 3—*

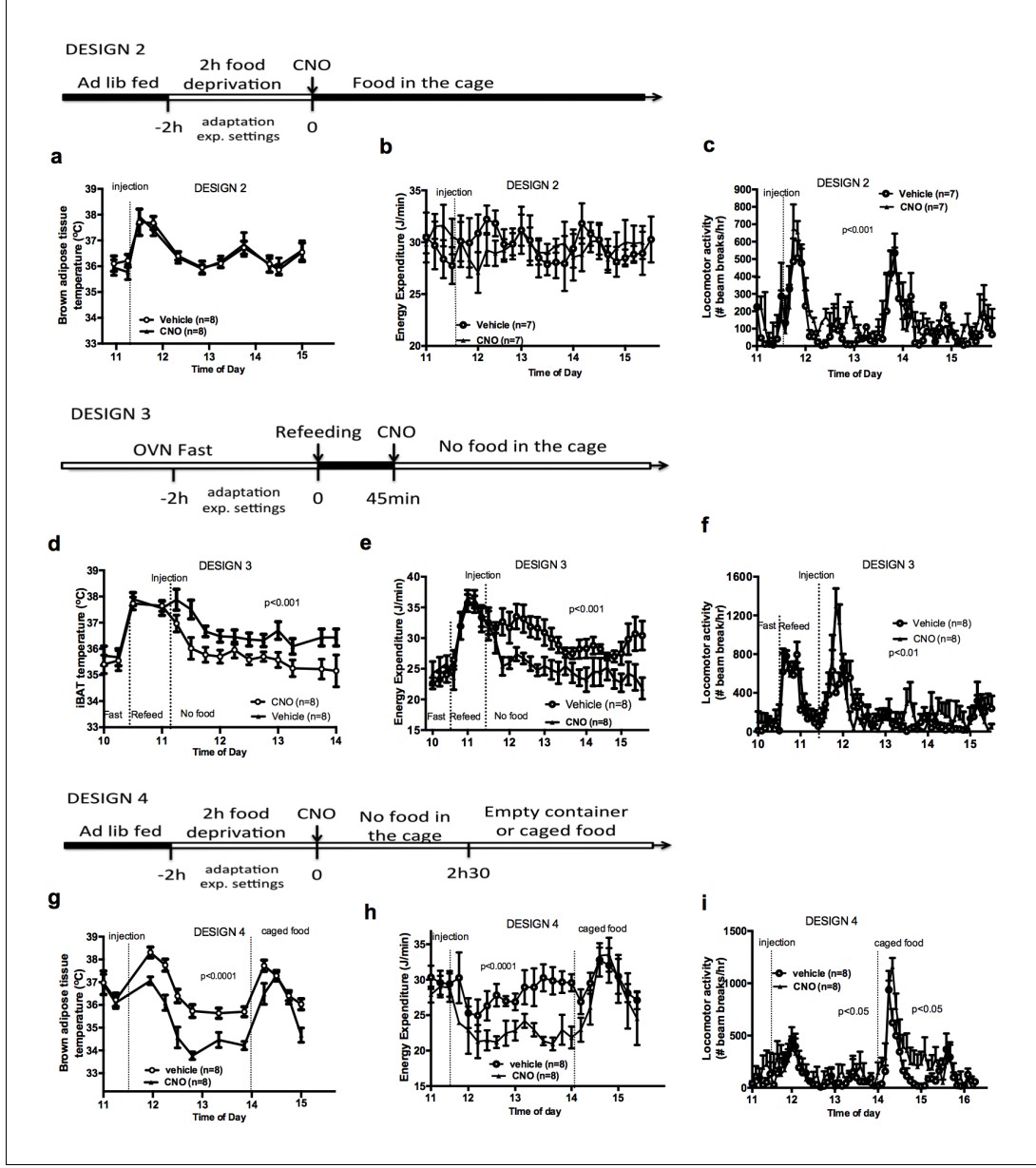

**Figure 2.** Food availability and sensory cues of food availability modulate the AGRP-iBAT circuit. iBAT temperature (**a**), energy expenditure (**b**) and locomotor activity (**c**) following chemogenetic activation of AGRP neurons in the presence of food (DESIGN 2). iBAT temperature (**d**), energy expenditure (**e**) and locomotor activity (**f**) following chemogenetic activation of AGRP neurons in refed mice (DESIGN 3). iBAT temperature (**g**), energy expenditure (**h**) and locomotor activity (**i**) following chemogenetic activation of AGRP neurons in the presence of caged food (DESIGN 4). Data are mean ± SEM, *p<0.05 ; ***p<0.001.

The following figure supplement is available for figure 2:

**Figure supplement 1.** Food availability and sensory cues of food availability modulate the AGRP-iBAT circuit.

figure supplement 1a and b). We used immunofluorescence to detect the active form of mTOR phosphorylated at Ser$^{2448}$ (p-mTOR) in hypothalamic slices. Rapamycin suppressed p-mTOR expression (*Figure 3—figure supplement 1c*), confirming the specificity of this staining. Refeeding for 1 hr following an overnight fast significantly increased p-mTOR expression in ARH NPY neurons (*Figure 3a and d*). Importantly, p-mTOR activation following refeeding occurs only in 15% of ARH

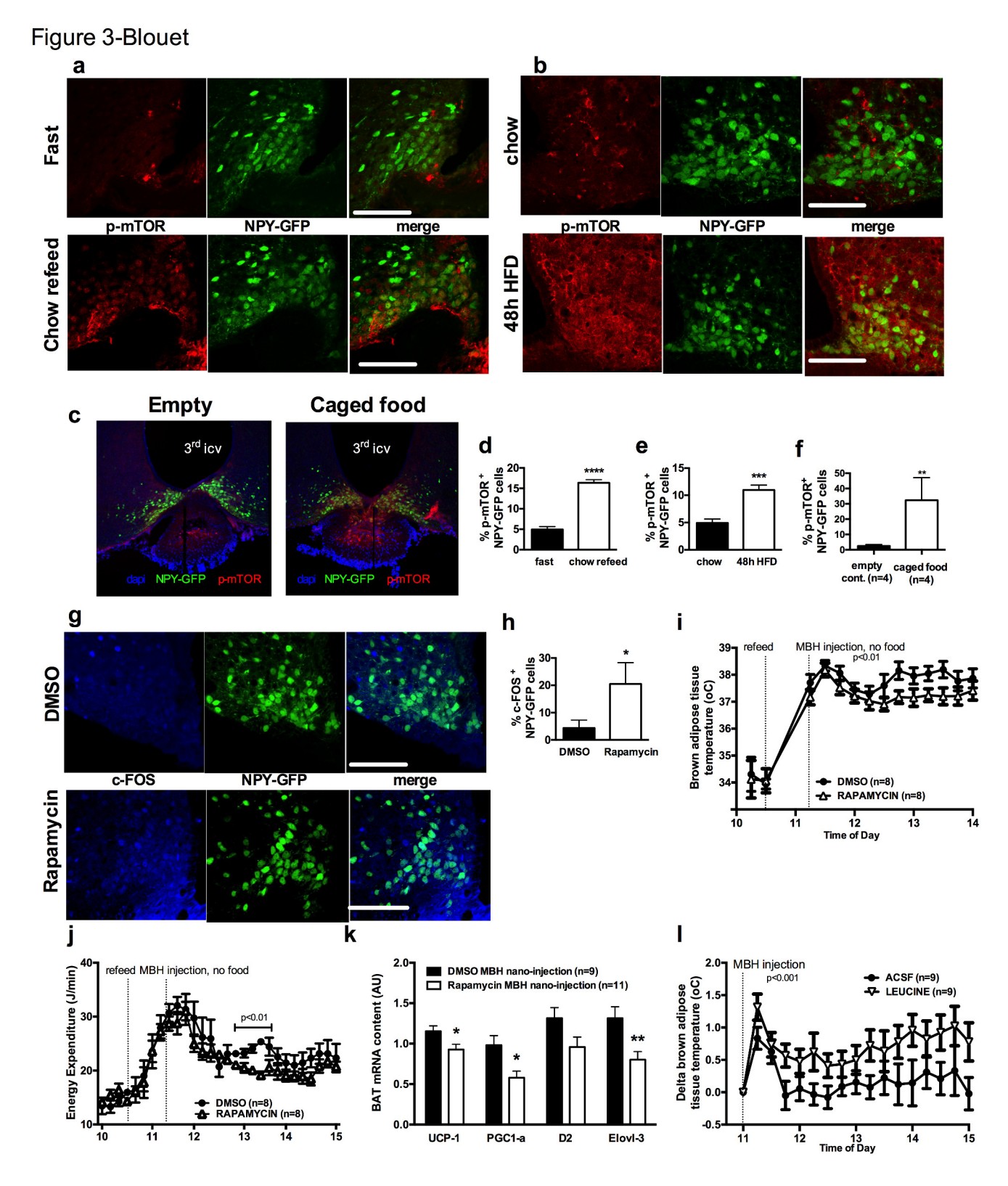

**Figure 3.** mTOR signaling in AGRP neurons senses endogenous and environmental signals of energy availability, regulates AGRP activity and iBAT thermogenesis. Activation levels of mTOR in *Npy*-GFP neurons following a 18 hr fast and 1 hr chow re-feed (a, d, n = 5), and a 48 hr transition from chow to HF diet (b, e, n = 6), and a 18 hr fast followed by a 30 min exposure to caged food (c, f). *Npy*-GFP neuronal c-fos expression following

*Figure 3 continued on next page*

Figure 3 continued

inhibition of mTORC1 using rapamycin (g, h) – scale bar is 100 μm. iBAT temperature (i), energy expenditure (j) and iBAT thermogenic gene expression (k) following MBH nanoinjection of rapamycin in refed mice. iBAT temperature (l) following MBH nanoinjection of L- leucine in fasted mice. Data are mean ± SEM *p<0.05 ; ***p<0.001 ; ****p<0.0001.
The following figure supplement is available for figure 3:

**Figure supplement 1.** mTOR signaling in AGRP neurons senses endogenous and environmental signals of energy availability, regulates AGRP activity and iBAT thermogenesis.

NPY/AGRP neurons, supporting the conclusion that postprandial changes in energy availability is coupled to changes in mTOR signaling only in a subpopulation of ARH NPY/AGRP neurons. Costained neurons were mostly absent from the rostral ARH (−1.22 to −1.6 mm post Bregma [*Paxinos and Franklin, 2001*]) and concentrated in the ventro-medial neurons of the caudal ARH (−1.7 mm to −2 mm post bregma [*Paxinos and Franklin, 2001*]) (*Figure 3—figure supplement 1d*).

Another metabolic context associated with a rapid activation of iBAT thermogenesis is exposure to a high fat (HF) diet (*Cannon and Nedergaard, 2004*). We confirmed that after 2 days of HF feeding, iBAT thermogenesis was increased (*Figure 3—figure supplement 1e and f*) leading to an increase in energy expenditure (*Figure 3—figure supplement 1g*), as previously reported (*Liu et al., 2014*). We observed that mTOR signaling was also upregulated in NPY/AGRP neurons in this context, again only in a small group of this neuronal population (*Figure 3b and e*). These data indicate that in nutritional conditions under which iBAT thermogenic activity is engaged, mTOR activity is increased in a subpopulation of AGRP/NPY neurons in which metabolic signals are coupled to changes in mTOR signaling. These findings raise the possibility that mTOR in AGRP neurons mediate changes in sympathetic tone to iBAT under these conditions.

Given our observations that the inhibition of iBAT thermogenesis induced by AGRP neuronal activation is abolished in the presence of food, we sought to assess whether hypothalamic mTOR signaling is also sensitive to food sensory stimuli. We first measured p-mTOR expression in NPY/AGRP neurons of overnight fasted mice exposed for 30 min to an empty contained or caged food. In control mice exposed to an empty container, p-mTOR was absent from ARH NPY-GFP neurons (*Figure 3c and f*). In contrast, 30% ARH NPY-GFP neurons expressed p-mTOR in response to 30 min exposure to caged food. These data indicate that mTOR signaling in AGRP neurons, in addition to responding to acute (refeeding) and chronic (48 hr HF feeding) internal signals of energy availability, also responds to environmental food-related cues.

## MBH mTORC1 signaling modulates AGRP neuronal activity and iBAT thermogenesis

To directly probe the link between mTOR activity and AGRP neuronal activity, we treated refed *Npy-GFP* mice with rapamycin, a well-characterized mTORC1 inhibitor, and quantified the expression of the marker of neuronal activation c-fos in *Npy* -GFP cells. Rapamycin increased the expression of cfos in *Npy* -GFP neurons (*Figure 3g and h*), supporting the interpretation that mTORC1 signaling negatively regulates the activity of AGRP/NPY neurons.

We then tested whether acute changes in mTORC1 activity in AGRP neurons modulate iBAT thermogenesis using nanoinjections of activators and inhibitors of the mTORC1 pathway into the MBH of wild-type mice. Rapamycin diluted in 100% DMSO was administered into the hypothalamic parenchyma in nanovolumes to acutely activate mTORC1 signaling. Rapamycin-mediated inhibition of MBH mTORC1 in refed mice significantly reduced iBAT temperature (*Figure 3i*), energy expenditure (*Figure 3j*), and altered iBAT thermogenic gene expression (*Figure 3k*) without significantly affecting locomotor activity (*Figure 3—figure supplement 1h*). Of note, although this DMSO concentration may induce cellular stress responses, the low volume bolus we injected did not produce noticeable changes in behavior, locomotor activity, energy intake and energy expenditure compared to non-injected mice under the same conditions (not shown). Conversely, activation of MBH mTORC1 using the amino acid L-leucine produced an increase in iBAT temperature in fasted mice (*Figure 3l*) but did not produce changes in energy expenditure (*Figure 3—figure supplement 1i*). Thus, acute

changes in MBH mTORC1 signaling are associated with bidirectional changes in iBAT thermogenesis.

## Constitutive activation of mTORC1 signaling in AGRP neurons increases iBAT thermogenesis and energy expenditure and protects against diet-induced obesity

To directly test the role of increased mTORC1 signaling specifically in AGRP neurons in the control of iBAT thermogenesis, we generated a lentivector expressing a constitutively active mutant of the mTORC1 effector p70 S6 kinase 1 (S6K1) in a cre-dependent manner (pCDH-CMV-FLEX-HA-S6K1-F5a). We delivered the pCDH-CMV-FLEX-HA-S6K1-F5a vector into the ARH of normally-developed adult *Agrp-IRES-cre* or WT mice (mice subsequently referred to as Agrp-CA-S6 and WT-CA-S6) using stereotactic nanoinjections. *Agrp-IRES-cre;Npy-gfp* were used to validate the construct. Ha staining indicated that over 95% of the cells expressing the construct were *Npy*-GFP neurons and that over 60% of *Npy*-GFP cells expressed the construct (*Figure 4—figure supplement 1a*). Increased expression of the active form of ribosomal protein S6 (phosphorylated at Ser 240–244, p-rpS6, main downstream effector of S6K) in ARH *Npy*-GFP neurons confirmed constitutive activation of the pathway in AGRP neurons of Agrp-CA-S6 mice compared to controls (*Figure 4—figure supplement 1b*).

Constitutive activation of S6K1 in AGRP neurons had no effect on food intake (*Figure 4a*), body weight gain (*Figure 4—figure supplement 1c*), adiposity (*Figure 4—figure supplement 1d*) or glucose tolerance (*Figure 4—figure supplement 1e*) when mice were maintained on a chow diet. Given the crucial role of AGRP neurons in feeding behavior and survival (*Luquet et al., 2005*), these results indicate that manipulation of mTORC1 signaling in AGRP neurons does not produce loss of AGRP function.

Upon exposure to a HF diet, Agrp-CA-S6 mice were protected against body weight gain (*Figure 4b*). We performed a detailed characterization of energy balance during the transition from chow to HF diet. Constitutive activation of S6K1 in AGRP neurons did not affect energy intake (*Figure 4a*) but produced a significant increase in night-time energy expenditure under HF diet (*Figure 4c*). The specific implication of energy expenditure in this metabolic phenotype, and the preservation of feeding behavior, support the conclusion that chronic changes in S6K1 signaling in AGRP neurons do not produce an overall change in AGRP activity, but rather modulate the activity of a specific subpopulation of AGRP neurons regulating energy expenditure. Substrate utilization was similar between groups on chow but fat utilization was substantially increased after the switch to the HF diet in Agrp-CA-S6 mice (*Figure 4d*). Likewise, locomotor activity significantly increased in Agrp-CA-S6 mice after the transition to the HF diet (*Figure 4e*). Night-time iBAT temperature was significantly increased in Agrp-CA-S6 mice (*Figure 4f*) and consistently, mRNA expression of iBAT thermogenic markers was increased (*Figure 4g*), suggesting a role for increased iBAT activation in the elevated energy expenditure and reduced weight gain observed in these mice. The amplitude of iBAT thermogenic response to $\beta$-adrenergic stimulation was similar between groups (*Figure 4h*). This indicates that elevated iBAT thermogenesis in *Agrp-IRES-cre* mice was not the result of an impairment in iBAT responsiveness to sympathetic activation. Instead, increased iBAT norepinephrine turnover in AGRP-CA-S6 mice compared to control indicated increased sympathetic tone to iBAT following constitutive activation of S6K in AGRP neurons (*Figure 4i*; NETO: −0.75 pM/mg/h vs. −1.26 pM/mg/h in WT-CA-S6 and AGRP-CA-S6 respectively, p<0.05). Constitutive activation of S6K1 in AGRP neurons also increased iBAT temperature (*Figure 4j*) energy expenditure (*Figure 4k*) during a fast resulting in enhanced body weight loss (*Figure 4l*). Postprandial increases in iBAT thermogenesis and energy expenditure were also higher in mice expressing constitutively active S6K1 in AGRP neurons (*Figure 4l and m*). Thus, constitutive activation of S6K signaling in AGRP neurons of adult normally developed mice protect against diet-induced weight gain through an increase in sympathetic tone to iBAT and an increase in iBAT thermogenesis.

## Inhibition of endogenous mTORC1 in AGRP neurons reduces iBAT thermogenesis and energy expenditure and promotes HF-induced weight gain

In order to investigate the role of endogenous mTORC1 signaling in AGRP neuronal control of energy homeostasis, we generated a lentivector expressing a kinase dead mutant of the S6K1 in a

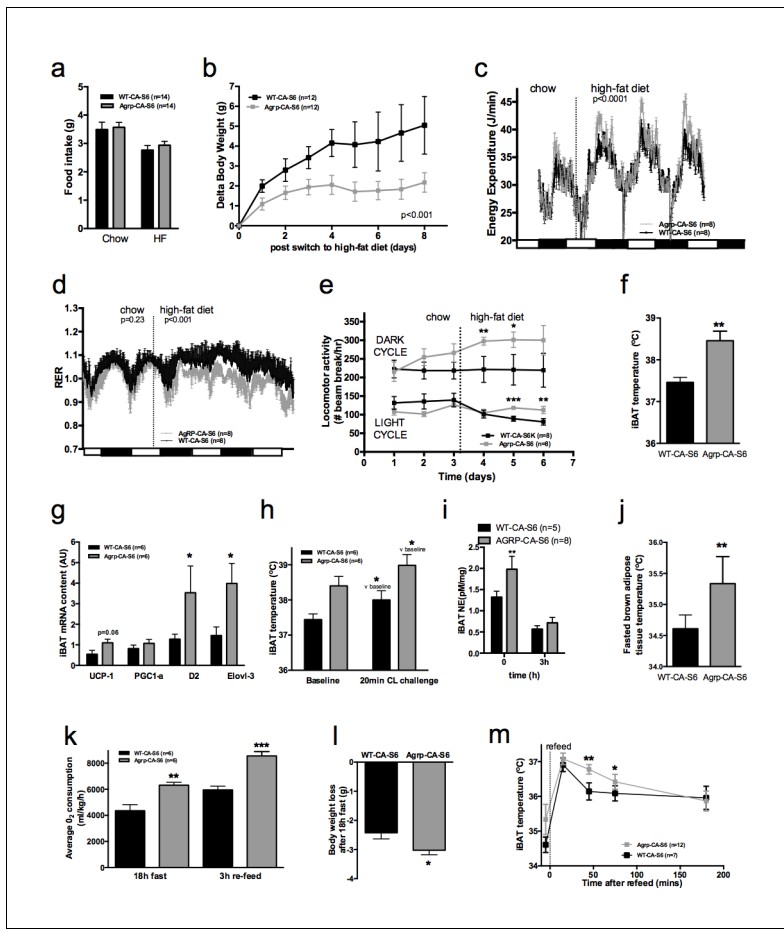

**Figure 4.** Metabolic phenotyping of mice expressing constitutively active mutant of mTORC1 effector p70 S6 kinase one in *Agrp-IRES-cre* (Agrp-CA-S6) or wild-type (WT-CA-S6) littermates. Food intake (**a**), body weight gain on high-fat diet (**b**), energy expenditure (**c**), RER (**d**), locomotor activity (**e**), iBAT temperature (**f**), iBAT thermogenic gene expression (**g**), iBAT temperature following β3-agonist-induced challenge (**h**), and Norepinephrine turnover (**i**) in Agrp-CA-S6 and WT-CA-S6 mice. iBAT temperature (**i**), energy expenditure (**j**), and body weight loss during 18 hr overnight fast (**k**), and iBAT temperature during 3 hr re-feed (**l**) in Agrp-CA-S6 and WT-CA-S6 mice. Data are mean ± SEM *p<0.05 ; **p<0.01 ; ***p<0.001.

The following figure supplement is available for figure 4:

**Figure supplement 1.** Metabolic phenotyping of mice expressing constitutively active mutant of mTORC1 effector p70 S6 kinase one in Agrp-IRES-cre (Agrp-CA-S6) or wild-type (WT-CA-S6) littermates.

cre-dependent manner (pCDH-CMV-FLEX-HA-S6K1-KR) and injected this virus into the MBH of normally developed adult *Agrp-IRES-cre* mice or wild-type littermates through local stereotatic nanoinjections (Agrp-DN-S6 and WT-DN-S6). *Agrp-IRES-cre;Npy-gfp* were used to validate the construct. Ha staining indicated that over 95% of the cells expressing the construct were *Npy*-GFP neurons and that over 40% of *Npy*-GFP cells expressed the construct (*Figure 5—figure supplement 1a*). Decreased expression of p-rpS6 in ARH *Npy*-GFP neurons confirmed inhibition of the pathway in AGRP neurons of Agrp-DN-S6 mice compared to controls (*Figure 5—figure supplement 1b*).

Quenching endogenous S6K1 in AGRP neurons produced changes in energy expenditure reciprocal to those observed following constitutive activation of the pathway. These were not sufficient to affect body weight gain during chow maintenance (*Figure 5—figure supplement 1c*) but produced a significant decrease in energy expenditure (*Figure 5a*), locomotor activity (*Figure 5b*), iBAT temperature (*Figure 5c*), and iBAT thermogenic gene expression (*Figure 5d*) after switch to a HF diet. Downregulation of S6K1 activity in AGRP neurons only produced a modest reduction in weight gain

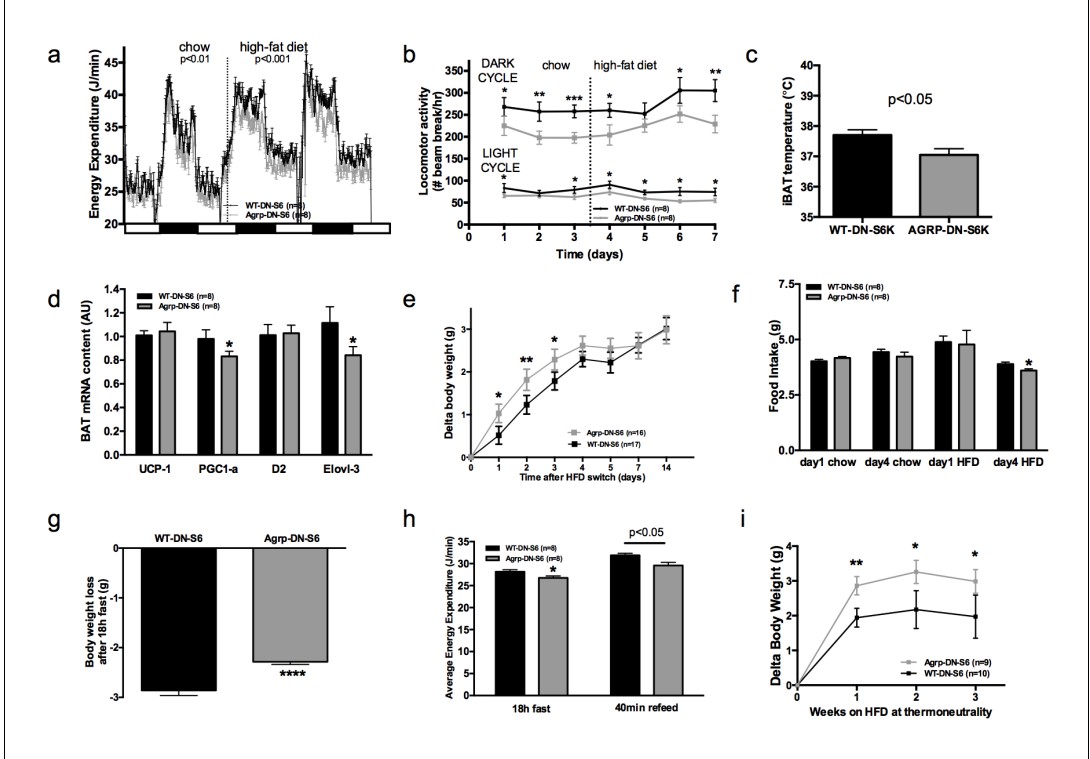

**Figure 5.** Metabolic phenotyping of mice expressing dominant negative mutant of mTORC1 effector p70 S6 kinase 1 (S6K1) in *Agrp-IRES-cre* (Agrp-DN-S6) or wild-type (WT-DN-S6). Energy expenditure (a), locomotor activity (b), iBAT temperature (c), iBAT thermogenic gene expression (d), body weight gain during initial switch to HF diet (e), food intake (f), body weight loss during overnight 18 hr fast (g), and energy expenditure during fast and re-feed (h) in Agrp-DN-S6 and WT-DN-S6 mice. Body weight gain of Agrp-DN-S6 and WT-DN-S6 mice at thermoneutrality under chow (i) and HF (k) diet. Data are mean ± SEM *p<0.05 ; **p<0.01; ***p<0.001.

The following figure supplement is available for figure 5:

**Figure supplement 1.** Metabolic phenotyping of mice expressing dominant negative mutant of mTORC1 effector p70 S6 kinase 1 (S6K1) in Agrp-IRES-cre(Agrp-DN-S6) or wild-type (WT-DN-S6).

during the initial switch to HFD (*Figure 5e*) during which time we did not observe any differences in food intake (*Figure 5f*) or RER (*Figure 5—figure supplement 1d*). However, we observed a reduction in food intake in DN-S6K mice 4 days after the introduction of the HF diet (*Figure 5f*), an effect that likely occurred secondary to reduced energy expenditure and mitigated changes in weight gain.

To further characterize energy expenditure without the confound of energy intake, we challenged these mice with an overnight fast. Despite showing no difference in body weight under *ad libitum* access to food, Agrp-DN-S6 mice lose significantly less weight than wild-type controls during an overnight fast (*Figure 5g*), indicating increased energy efficiency in these mice. Consistently, energy expenditure was lower in Agrp-DN-S6 mice throughout the fast (*Figure 5h*). In addition, postprandial increases in energy expenditure (*Figure 5h*) during the subsequent refeed was lower. Last, body weight gain at thermoneutrality was significantly higher in Agrp-DN-S6 mice than in controls (*Figure 5i*). Collectively, these data indicate that downregulation of S6K signaling in AGRP neurons leads to a decrease in energy expenditure mediated at least in part by a decrease in iBAT thermogenesis, which leads to a significant increase in body weight gain when mice are maintained at thermoneutrality.

# mTORC1 in AGRP neurons mediates the thermogenic effect of leptin

Central leptin engages the sympathetic nervous system to increase iBAT thermogenesis through the activation of distributed leptin-receptor expressing neurons in the DMH, nMPO, NTS and ARH. Several lines of evidence support a role for leptin receptors in the ARH in leptin-induced activation of iBAT (*Rahmouni and Morgan, 2007*; *Harlan et al., 2011*). In the adult brain, leptin inhibits the vast majority of AGRP neurons (*Baquero et al., 2014*). However, leptin-induced hyperpolarization of AGRP neurons has not been implicated in leptin-induced increase in sympathetic tone to iBAT. Previous work implicated hypothalamic mTORC1 signaling in leptin's effects on energy balance (*Cota et al., 2006*; *Blouet et al., 2008*). Thus, we hypothesized that leptin activates mTORC1 signaling in AGRP neurons, leading to an increase in sympathetic tone to iBAT.

To test this hypothesis, we first colocalized markers of mTORC1 signaling in NPY/AGRP neurons 30 min following leptin administration. Leptin increased the expression of p-mTOR (*Figure 6a and c*) and the expression of the active form of one of its effectors, ribosomal protein S6 (p-rpS6, *Figure 6b and d*) in NPY/AGRP neurons. As expected, leptin increased expression of pSTAT3 in NPY/AGRP neurons, and we observed that 40% of pSTAT-3+ NPY/AGRP neurons also expressed the active form of rpS6.

We then measured leptin's thermogenic effect in mice with bidirectional modulations of S6K1 signaling in AGRP neurons. Leptin's effects on temperature were attenuated in Agrp-CA-S6 mice compared to wild-type controls (*Figure 6e*). While both groups reached a similar absolute temperature increase following ip leptin, the relative change in temperature was significantly lower in the Agrp-CA-S6 mice due to their elevated core temperature at baseline. Together these data suggest that lack of flexibility in mTORC1 signaling in AGRP neurons impairs leptin's thermogenic action. Leptin's thermogenic effect was completely abolished in Agrp-DN-S6 mice (*Figure 6f*), indicating that S6K1 signaling in AGRP neurons is required for leptin's thermogenic action.

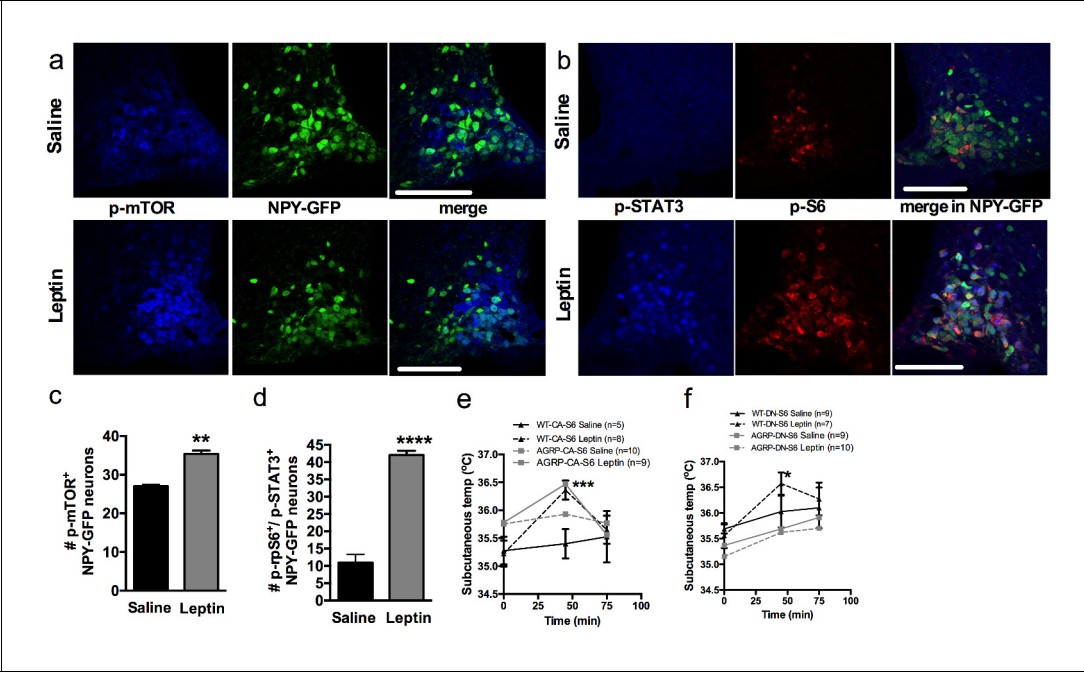

**Figure 6.** Leptin engages mTORC1 signaling in AGRP neurons to regulate thermogenesis. Expression of p-mTORC1 (a, c) and colocalization of p-rpS6 and p-STAT3 (b, d) in *Npy*-GFP neurons following systemic leptin treatment. Leptin's thermogenic effects in WT-CA-S6 and Agrp-CA-S6 (e), and WT-DN-S6 and Agrp-DN-S6 (f) mice. Data are mean ± SEM *p<0.05 ; **p<0.01 ; ***p<0.001.

## Discussion

Activation of AGRP neurons has been established as an essential component in the initiation of feeding behavior (*Aponte et al., 2011*; *Krashes et al., 2011*). Here, we demonstrate that the multi-modal contribution of AGRP neurons to the maintenance of energy balance includes the regulation of iBAT thermogenesis. We show that activation of AGRP neurons induces a rapid suppression of sympathetic output to iBAT, as previously reported (*Steculorum et al., 2016*), leading to a decrease in iBAT thermogenesis and energy expenditure. We show that the AGRP-iBAT circuit is selectively engaged to spare internal energy stores in the absence of food and food sensory cues, supporting the notion that AGRP neurons integrate external and internal signals of energy availability to coordinate energy intake and expenditure in conditions of low energy availability. We provide evidence that this regulation also occurs at thermoneutrality and is therefore potentially relevant to human physiology. We identify the mTORC1 signaling pathway as an intracellular mechanism through which AGRP neurons monitor external and internal energy availability, and demonstrate that mTORC1 signaling modulates AGRP activity, sympathetic tone to iBAT, iBAT thermogenesis and energy expenditure. Last, we show that metabolic sensing via mTORC1 in AGRP neurons is required for the regulation of energy expenditure both during nutritional transitions from fasting to feeding and during adaptation to HF feeding. Our data provide a physiological framework for the role of AGRP neurons in the control of adaptive thermogenesis and the coordination of energy intake and energy expenditure, and represent the first characterization of the functional consequences of AGRP sensory integration in the regulation of energy balance.

A previous study implicated AGRP neurons in the regulation of energy expenditure (*Krashes et al., 2011*). In that study, chemogenetic activation of AGRP neurons produced a rapid reduction in oxygen consumption. Although the effector mediating this decrease had not been investigated, these results are in line with our findings and support a role for AGRP neurons in the regulation of energy expenditure. However, the reduction in oxygen consumption occurred in the presence of food, which conflicts with our findings but may result from differences in experimental conditions. In our experimental paradigm, animals were all adapted to the experimental environment for 2 hr in the morning before CNO administration, a period during which they had no access to food or food-related sensory cues. We restored access to food immediately after CNO administration, whereas in the Krashes paper, food was present before and after the injection. This suggests that changes in environmental food availability via food presentation are necessary to inhibit the AGRP-iBAT circuit, and that environmental food cues do not maintain a tonic inhibition of AGRP neurons. This interpretation is in fact consistent with our observation that caged-food only transiently blunts AGRP-induced suppression of energy expenditure and iBAT temperature.

In a study using capsaicin-dependent chemogenetic activation of AGRP neurons, activation of AGRP neurons produced a modest and transient decrease in energy expenditure that was attributed to a rapid suppression of inguinal WAT (iWAT) browning (*Ruan et al., 2014*). These authors ruled out a role for iBAT in this effect based on the lack of change in iBAT thermogenic gene expression (*Ruan et al., 2014*). This latter observation is consistent with (*Steculorum et al., 2016*), our results, and a reported half-life of 30–72 hr of UCP1 protein in iBAT (*Clarke et al., 2012*). Nonetheless, our data indicate that iBAT temperature can rapidly decrease following AGRP neuronal activation through activation of sympathetic signaling to iBAT and decreased activation of HSL, an enzyme implicated in fatty acid mobilization and heat production from iBAT. We can not exclude that a decrease in UCP1 expression in iWAT may contribute to the rapid decrease in energy expenditure observed following AGRP neuronal activation. However, we report an average decrease in energy expenditure of 8 J/min at ambient temperature (25% decrease compare to controls and baseline levels). While iBAT thermogenesis can account for this portion of energy expenditure in mice housed at ambient temperature (*Cannon and Nedergaard, 2011*), current knowledge does not indicate that it is the case for beige adipocyte (*Nedergaard and Cannon, 2013*) further supporting a major role of iBAT inhibition in the decrease in energy expenditure induced by AGRP neuronal activation.

Recently, AGRP neurons have been implicated in the regulation of insulin sensitivity via iBAT myostatin expression (*Steculorum et al., 2016*). This work extends the orchestration of the metabolic adaptations to energy deficit by AGRP neurons to include a suppression of peripheral glucose uptake, which may serve as a glucose sparing process to protect the brain from hypoglycemia in conditions of low energy availability.

The neurochemistry and neuroanatomical circuits underpinning AGRP neuronal regulation of iBAT remain to be fully characterized. Circuits downstream from AGRP neurons previously implicated in the modulation iBAT thermogenesis include NPYergic (*Shi et al., 2013*), melanocortinergic (*Small et al., 2003*; *Berglund et al., 2014*; *Brito et al., 2007*), and GABA-ergic circuits (*Tong et al., 2008*). This diversity may underlie the coexistence of multiple circuits engaged under different metabolic situations or within different time frames (*Krashes et al., 2011*), to adjust iBAT thermogenic function in response to changes in AGRP tone. Consistent with the idea that multiple circuits may contribute to the AGRP-iBAT axis, AGRP projections to the LH and the anterior bed nucleus of the stria terminalis have both been shown to mediate the effect of AGRP neurons on iBAT myostatin expression (*Steculorum et al., 2016*). Interestingly, although the paraventricular nucleus of the hypothalamus (PVH) receives dense axonal input from AGRP neurons (*Haskell-Luevano et al., 1999*) and has been implicated in the regulation of sympathetic output to iBAT (*Kong et al., 2012*; *Morrison and Madden, 2014*), AGRP projections to the PVH do not modulate iBAT myostatin expression. While a large proportion of PVH neurons negatively regulate iBAT thermogenesis (*Kong et al., 2012*; *Madden and Morrison, 2009*), PVH oxytocin neurons positively regulate sympathetic tone to iBAT (*Morrison and Madden, 2014*) and have been implicated in diet-induced thermogenesis (*Wu et al., 2012*). However, whether AGRP neurons project to PVH oxytocin neurons is not clear (*Atasoy et al., 2014*; *Garfield et al., 2015*). PVH TH neurons represent an alternative population that may be involved in the AGRP-iBAT axis, as tonic inhibition from ARH NPY neurons to PVH TH neurons via Y1R signaling has also recently been implicated in the control of iBAT thermogenesis (*Shi et al., 2013*). DMH neurons may alternatively or concomitantly mediate the effect of AGRP neuronal activation on iBAT thermogenesis, as the DMH is a major source of sympathoexcitatory input to medullary iBAT sympathetic premotor neurons in rRPa (*Morrison and Madden, 2014*), and GABAergic inhibitory tone to the DMH suppresses iBAT thermogenesis (*Cao et al., 2004*), raising the possibility that iBAT-regulating GABAergic input to the DMH originates from AGRP neurons.

Ghrelin is a well-established endogenous activator of AGRP neurons and the ghrelin receptor is predominantly expressed in AGRP neurons in the ARH (*Willesen et al., 1999*). Our observation that parenchymal administration of ghrelin in the ARH reproduces the effects of AGRP neuronal activation on iBAT temperature supports the physiological relevance of the AGRP-iBAT circuit. While the contribution of iBAT thermogenesis to the metabolic effects of ghrelin has been previously reported (*Tsubone et al., 2005*; *Yasuda et al., 2003*), the neuroanatomical sites responsible for this effect had not been identified. Together with the published literature implicating AGRP neurons in the orexigenic effects of ghrelin (*Wang et al., 2014*; *Andrews et al., 2008*), our data lend strength to the idea that AGRP neurons coordinate the feeding and metabolic effects of ghrelin.

Our findings establish a role for mTORC1 signaling in AGRP neurons in the detection of sensory and metabolic signals of energy availability and the regulation of the AGRP-iBAT axis. These findings may seem conflicting with a previous report showing increased ribosomal protein S6 (rpS6) signaling in AGRP neurons of fasted mice (*Villanueva et al., 2009*). In fact, rpS6 signaling receives activatory inputs from many neuronal signaling pathways, is now acknowledged as a marker of neuronal activation (*Knight et al., 2012*) and therefore represents an inappropriate surrogate marker of mTORC1 signaling in this context. Our data support instead that mTORC1 activation in AGRP neurons negatively regulates AGRP activity. Furthermore, we provide evidence that increased mTORC1 signaling in AGRP neurons contributes to diet-induced thermogenesis during early exposure to HF feeding and protects against diet-induced obesity (DIO). Consistent with these findings, acute HF feeding has been shown to increase S6K1 activity (*Cota et al., 2008*) and reduce AGRP signaling (*Ziotopoulou et al., 2000*).

While there is convincing evidence that acute and chronic energy excess result in increased energy expenditure in rodents (*Cannon and Nedergaard, 2004*) and humans (*Ravussin et al., 2014*; *Vosselman et al., 2013*), the nature of the signals mediating these effects is unclear and may include specific nutrients and/or caloric load per se. Interestingly, the lack of effect of increased mTORC1 signaling in AGRP neurons under chow maintenance reveals that activation of this pathway is not sufficient to modulate energy balance, and suggests that its contribution to energy expenditure is contingent on energy surfeit. Controversy persists on the role of leptin during energy surfeit in diet-induced hypermetabolism humans (*Heymsfield et al., 1999*; *Mackintosh and Hirsch, 2001*), and recent data revealed novel mechanisms through which leptin modulates core temperature independently of energy expenditure (*Fischer et al., 2016*). Nevertheless, in rodents, leptin increases

sympathetic output to iBAT, oxygen consumption and iBAT temperature (*Morrison, 2004*; *Enriori et al., 2011*; *Rahmouni and Morgan, 2007*), and neuroanatomical and functional studies indicate that activation of central leptinergic circuits can rapidly increase iBAT thermogenesis (*Yu et al., 2016*). Our study identifies AGRP neurons as a novel neurochemical population implicated in leptin's thermogenic action, although they do not directly implicate iBAT thermogenesis in the thermogenic response to leptin.

Previous work investigating the role of hypothalamic mTORC1 signaling support the prediction that modulation of S6K1 in AGRP neurons would affect food intake (*Cota et al., 2006*; *Blouet et al., 2008*). The failure to observe any effect on feeding in our different experimental paradigms indicates that chronic changes in mTORC1 signaling in AGRP neurons are not sufficient to alter energy intake. This does not preclude a contribution of this pathway in the acute control of feeding behavior. In fact, we observe that rapamycin rapidly activates AGRP neurons in refed mice, and under similar conditions hypothalamic rapamycin has been shown to rapidly increase foraging and food intake (*Cota et al., 2006*). Here, we demonstrate that rapamycin selectively suppresses iBAT temperature and energy expenditure in the absence of food, precisely mirroring the metabolic effects of AGRP neuronal activation. Interestingly, we observed that only a portion of AGRP neurons expresses active mTORC1 following a refeed or 2 days of HF feeding, indicating that the mTORC1 pathway is nutritionally regulated selectively in a subset of this neuronal population. Our findings contrast with other models of altered mTORC1 signaling in AGRP neurons, in which neonatal deletion of S6K1 or raptor in AGRP neurons failed to affect energy balance (*Smith et al., 2015*; *Albert et al., 2015*). Importantly, in contrast to the aforementioned studies, our model identifies the role of mTORC1 signaling in AGRP neurons in the physiology of mice with normally-developed hypothalamic circuits.

Altogether, our studies uncover a role for mTORC1 signaling within AGRP neurons in surveying energy availability to engage iBAT thermogenesis and identify AGRP neurons as a neuronal substrate for the coordination of energy intake and adaptive expenditure under various physiological contexts.

## Materials and methods

### Animals

All mice were group-housed unless otherwise stated and maintained in individually ventilated cages with standard bedding and enrichment. Mice were maintained in a temperature and humidity-controlled room on a 12 hr light/dark cycle with *ad libitum* access to water and standard laboratory chow diet or HF diet D12266B, 31.8% kcal as fat, 4.4 kcal/g, Research Diets) unless otherwise stated. *Agrp-IRES-cre* mice were genotyped with the following primers: 5′-GGGCCCTAAGTTGAGTTTTCCT-3′; 5′-GATTACCCAACCTGGGCAGAAC-3′ and 5′-GGGTCGCTACAGACGTTGTTTG-3′. All experiments were carried on heterozygous *Agrp-IRES-cre* mice and their wild-type littermates. *Npy-GFP* mice were genotyped with the following primers: 5′-TATGTGGACGGGGCAGAAGATCCAGG-3′; 5′-CCCAGCTCACAT ATTTATCTAGAG-3′; 5′-GGTGCGGTTGCCGTACTGGA-3′. *Npy-GFP/Agrp-IRES-Cre* mice were obtained by crossing homozygous *Npy*-GFP mice with homozygous Agrp-IRES-Cre mice. All experiments were performed in accordance with the Animals (Scientific Procedures) Act 1986 and approved by the local animal ethic committees.

### Surgical procedures

Surgical procedures were performed under isofluorane anesthesia, and all animals received Metacam prior to the surgery, 24 hr after surgery and were allowed a 1 week recovery period during which they were acclimatized to injection procedures. Mice were stereotactically implanted with bilateral steel guide cannulae (Plastics One) positioned 1 mm above the ARH (A/P: −1.1 mm, D/V: −4.9 mm, lateral: +0.4 mm from Bregma), as previously described (*Blouet et al., 2012*). Beveled stainless steel injectors (33 gauge) extending 1 mm from the tip of the guide were used for injections. All viral brain injections were performed on 11 week old mice at 100 nl/min, 500 nl/side (lentivectors, 1–2 $\times$ 10$^9$ pfu/ml) and 200 nl/side (AAV-hSyn-DIO-hM3D(Gq)-mCherry, UNC Vector Core, 1 $\times$ 10$^{12}$ pfu/ml). For chronic cannulae implantation, cannula guide was secured in place with Loctite glue and dental cement (Fujicem). Correct targeting was confirmed histologically postmortem (placement of cannula guide track) or using mCherry immunofluorescent staining. Temperature telemetric probe (IPTT-300,

BMDS) were inserted subcutaneously or secured below the interscapular brown adipose pad. All subsequent functional studies were performed in a crossover randomized manner on age- and weight-matched groups after 1 week recovery and 1 week daily acclimatization, and at least 4 days elapsed between each injection.

## Preparation of lentiviral plasmids

pRK7-HA-F5A (Addgene 8986) and pRK7-HA-KR (Addgene 8985) were a gift from John Blenis (*Schalm and Blenis, 2002*). pRK7-HA-F5A and pRK7-HA-KR were digested with XbaI followed by a fill-in reaction for blunt end ligation and a digestion with EcoRI to isolate the HA-F5A and HA-KR constructs. A lentiviral construct previously developed to express hTXNIP in a cre-dependent manner (*Blouet et al., 2012*), pCDH-CMV-FLEX-hTXNIP, was digested with Xho-1 followed by a fill-in reaction for blunt end ligation and a digestion with EcoR1 to isolate the pCDH-CMV-FLEX vector. The two inserts were ligated to the pCDH-CMV-FLEX lentivector to produce pCDH-CMV-FLEX-Ha-S6K1-F5A and pCDH-CMV-FLEX-Ha-S6K1-KR. Several clones were screened using One Shot Stbl3 E Coli competent cells (ThermoFisher and verified by sequencing. Plasmids were subsequently used for packaging reactions to generate viral stocks suitable for transfection by System Bioscience (Mountain View, CA). Cre-dependent expression of S6K1 constructs was confirmed in HEK293T cells (ATCC, negative for mycoplasma contamination) transfected with a pCAG-Cre-IRES2-GFP plasmid, a gift from Dr. Anjen Chenn's lab (Addgene 26646).

## Metabolic phenotyping

### AGRP chemogenetic activation

On the day of the procedure, mice housed in their home cage were acclimatized to the procedure room and food deprived during the 2 hr preceding the injections. Mice received CNO (Sigma, 1 mg/kg) or vehicle and were maintained in a food-free cage for the four following hours unless otherwise stated. iBAT, intraperitoneal or subcutaneous temperature were measured using remote biotelemetry (IPTT-300 Bio Medic Data Systems). Energy expenditure was assessed using indirect calorimetry in a Metabolic-Trace system (Meta-Trace, Ideas Studio, UK).

### Brain injections

For ghrelin or leucine injection studies, mice housed in their home cage were acclimatized to the procedure room and food deprived during the 2 hr preceding the injections. Mice received bilateral parenchymal nanoinjection of ghrelin (Phoenix Pharmaceuticals, 1 mg/ml, 100 nl/side, 100 nl/min), L-leucine (Sigma, 2.1 mM, 100 nl/side, 100 nl/min) or aCSF (R and D), were immediately returned to their home cage, and iBAT temperature and energy expenditure were monitored as described above. For rapamycin injection studies, mice were fasted overnight, re-fed for 1 hr in the morning, nanoinjected with rapamycin or DMSO (Merck Millipore, 5 mM, 150 nl/side), placed in clean, food-free cages, and iBAT temperature and energy expenditure were monitored as described above.

### Oral glucose tolerance

Glucose tolerance was measured on body-weight matched animals. Mice were food deprived for 6 hr and blood was sampled from tail vein immediately prior to glucose bolus (gavage, 1 mg/kg), and 15, 30, 60 and 120 min following bolus administration. Blood glucose was analyzed using a glucometer (Precision Xtra; MediSense).

### Body composition analysis

Body composition was determined by magnetic resonance spectroscopy (MRS) using an Echo MRS instrument (Echo Medical Systems).

### CL challenge

After blood collection for basal measurements, mice received an intraperitoneal injection of 1 mg/kg BW CL316243 (5-[(2R)-2-[[(2R)-2-(3-chlorophenyl)-2-hydroxyethyl]amino]propyl]-1,3-benzodioxole-2,2-dicarboxylic acid disodium salt; Sigma), a $\beta$3 adrenergic agonist. Brown fat temperature was monitored as described above, over the 60 min after the CL316243 injection, and brown fat temperature change over that time period was calculated.

### Leptin challenge

Mice received an intraperitoneal injection of leptin (5 mg/kg, R&D) or saline 1 hr before the onset of the dark, and iBAT temperature, 24 hr body weight gain and 24 hr food intake were measured.

## RT-qPCR

qPCR was carried out as previously described (*Burke and Heisler, 2015*). Total RNA was purified from interscapular iBAT using RNA STAT 60 (AMS Biotechnology, Abington, UK) according to the manufacturer's instructions. cDNA was obtained by reverse transcription of 500 ng iBAT RNA. PCR of cDNA was performed in duplicate on an ABI Prism 7900 sequence detection system using Taqman Gene expression assay for *Elovl3*, *Pgc-1a*, *Ucp-1* and *Dio2* (Supplement 1). Data expressed as arbitrary units and expression of target genes corrected to the geometric average of four housekeeping genes: *18s*, *36β4*, *βactin* and *Gapdh*.

## Western blots

Tissues were ground to a fine powder using a sterile pestle and mortar on liquid nitrogen. Powdered tissue were resuspended in lysis buffer (50 mM Tris-HCL, 150 mM NaCl, 1 mM EGTA, 1 mM EDTA, 10 mM glycerophosphate, 2 mM orthovanadate, 2 mM PMSF,. 5% sodium deoxycholate, 1% Triton X-100, pH7.5) with added protease and phosphatase inhibitor cocktails according to manufacturer's instruction (Roche). Lysates were cleared by centrifugation at 2600 g for 5 min at 4°C, to separate the fat content of the samples. Protein extracts were further cleared by centrifugation at 10,000 *g* for 10 min at 4°C. Protein concentrations of the supernatants were determined using the Bradford assay, proteins were diluted in Laemli buffer and separated by SDS–polyacrylamide gel (12%) electrophoresis and transferred to Immobilon-P (Millipore) membrane). Membranes were blocked for 1 hr at room temperature and incubated overnight at 4°C with the indicated antibody (1:1000, Cell Signaling Technology). Bound primary antibodies were detected using peroxidase-coupled secondary antibodies and enhanced chemiluminescence (Amersham). Relative quantification of band intensities was calculated by digitally photographing exposed films and using Genesnap and Genetools software (Syngene).

## Brain perfusion, immunohistochemistry, confocal microscopy and image analysis

Animals were anaesthetized with Euthatal solution 80 mg/kg in saline and transcardiacally perfused with 4% paraformaldehyde. Brains were extracted and post-fixed in 4% paraformaldehyde, 30% sucrose for 48 hr at 4°C. Brains were sectioned using a freezing sliding microtome into 5 subsets of 25 microns sections. Antigen retrieval was used for all experiments prior to antibody incubation. Sections were incubated in 10 mM sodium citrate at 80°C for 20 min then washed three times in PBS. To decrease endogenous autofluorescence, samples were then treated with 0.3% glycine and 0.003% SDS in methanol. Tissue was blocked for 1 hr with 5% normal donkey serum at room temperature, and incubated at 4°C with primary antibodies against pSer2448-mTORC1 (72 hr, 1:65; Cell Signaling Technology), c-fos (48 hr, 1:5000, Synaptic Systems), pSer240/244rpS6 (19 hr, 1:200, Cell Signaling Technology), dsRed (1:1000, Clontech), or pSTAT3 (19 hr, 1:500, Cell Signaling Technology). Sections were then mounted on slides and coverslipped with Vectashield (Vector).

Sections were imaged using a Zeiss LSM510 confocal microscope with the 40x objective. Gain and laser power settings remained the same between experimental and control conditions.

Images of tissue sections were digitized, and areas of interest were outlined based on cellular morphology. The brain region evaluated was arcuate nucleus (1.5–1.9 mm caudal, 0–0.45 mm mediolateral, and 5.6–6 mm ventromedial to bregma), corresponding to the coordinates in the brain atlas of *Paxinos and Franklin, 2001*. Images were analyzed using the ImageJ manual cell counter.

## Serum analysis

Mouse FGF-21 was measured using a Quantikine ELISA kit from Bio-Techne (MF2100). The assay was performed according to the manufacturer's instructions. Samples were analyzed in duplicate and the mean value reported. Quality control samples were analyzed at the beginning and end of the batch to ensure consistency across the plate.

T4 was measured using a fully automated assay on the Siemens Dimension EXL analyzer. The method is an adaptation of the EMIT homogenous immunoassay technology. T4 is stripped from its binding proteins using a releasing agent. The released T4 binds to anti-T4 antibody, thereby reducing the amount of T4-glucose-6-phosphate dehydrogenase conjugate (T4-G6PDH) that can be bound to the antibody. The T4-G6PDH which is not bound to the antibody catalyzes the oxidation of glucose-6-phosphate with the simultaneous reduction of $NAD^+$ to NADH which absorbs at 340 nm. In the absence of T4, T4-G6PDH binds with the antibody and enzyme activity is reduced. The increase in absorbance at 340 nm due to the formation of NADH over a 60 s measure period is proportional to the activity of the T4-G6PDH. The T4 concentration is measured using a bichromatic (340, 383 nm) rate technique. All reagents and calibrators were purchased from Siemens. Samples were analyzed in singleton. Three levels of Quality Control samples were analyzed at the beginning and end of the batch. All QC results were within their target ranges before the sample results were reported.

## iBAT norepinephrine turnover

iBAT norepinephrine turnover was determined based on the fall in iBAT norepinephrine content following inhibition of catecholamine biosynthesis with α-methyl-DL-tyrosine methyl ester hydrochloride (AMPT,Sigma-Aldrich) administered ip at 250 mg/kg, as described previously (*Brodie et al., 1966*). AMPT is a competitive inhibitor of tyrosine hydroxylase, the rate-limiting enzyme in catecholamine biosynthesis. After AMPT administration, the endogenous tissue levels of norepinephrine decline and the slope of this decrease in tissue norepinephrine content is multiplied by the initial norepinephrine concentration to yield NETO, an indirect readout of SNS outflow.

Norepinephrine was assayed in iBAT using reversed phase high performance liquid chromatography (HPLC) and electrochemical detection (*Dalley et al., 2002*). Samples (20 mg) were homogenized in 0.2 ml of 0.2M perchloric acid and centrifuged at 8000 rpm for 20 min at 10°C. The supernatant was diluted 1:4 in PCA buffer before the injection. Twenty-five µl aliquots of supernatant were injected onto a C18 ODS 3 µm column (100 mm length x 4.6 mm i.d.,Hypersil Elite, Phenomenex, UK) with a mobile phase consisting of citric acid (31.9 g/L), sodium acetate (2 g/L), octanesulfonic acid (460 mg/L) EDTA (30 mg/L) and 15% methanol (pH 3.6). Norepinephrine was detected using an ESA Coulochem II detector with electrode 1 held at −200 mV and electrode two held at +250 mV. Chromatograms were acquired and analyzed using Chromeleon software (Dionex, UK).

## Statistical analysis

All data, presented as means ± SEM, have been analyzed using GraphPad Prism 6. For all statistical tests, an α risk of 5% was used. All kinetics were analyzed using repeated-measures two-way ANOVAs and adjusted with Bonferroni's post hoc tests. Multiple comparisons were tested with one-way ANOVAs and adjusted with Tukey's post hoc tests. Single comparisons were made using one-tail Student's t tests.

## Acknowledgements

This work was supported by the Medical Research Council New Blood Fellowship [MR/M501736/1] to CB, a NIDDK K99/R00 award to CB, the Medical Research Council Metabolic Disease Unit programme grant, the Disease Model Core facilities, and the Wellcome Trust Cambridge Mouse Biochemistry Laboratory. Animal procedures were performed under Tony Coll's home office PPL 80/2497 and Toni Vidal-Puig PPL 80/2484.

## Additional information

### Funding

| Funder | Grant reference number | Author |
| --- | --- | --- |
| Medical Research Council | MR/M501736/1 | Clémence Blouet |
| National Institutes of Health | | Clémence Blouet |

The funders had no role in study design, data collection and interpretation, or the decision to submit the work for publication.

## Author contributions

LKB, Data curation, Formal analysis, Investigation, Writing—original draft; TD, JM, Data curation, Formal analysis, Investigation, Methodology; ARC, Data curation, Formal analysis, Investigation; SV, Conceptualization, Validation, Methodology, Writing—original draft; ER, Investigation, Methodology; S-ML, JWD, Resources, Methodology; JX, Resources, Data curation, Formal analysis, Investigation, Methodology; KB, Resources, Data curation, Formal analysis, Methodology; SC, Resources, Validation, Methodology; TV-P, Conceptualization, Resources, Writing—original draft; GJS, Conceptualization, Writing—original draft, Writing—review and editing; CB, Conceptualization, Resources, Data curation, Formal analysis, Supervision, Funding acquisition, Validation, Investigation, Visualization, Methodology, Writing—original draft, Project administration, Writing—review and editing

## Author ORCIDs

Emma Roth, http://orcid.org/0000-0001-7790-7274
Clémence Blouet, http://orcid.org/0000-0002-1752-1270

## Ethics

Animal experimentation: This study was performed in strict accordance with the United Kingdom Home Office legislation (Animal (Scientific Procedure) Act 1986). The protocols were approved by the Animal Welfare and Ethics Review Committee and the Named Animal Care and Welfare Officer (study plans 24840114, 24840214, 24840314, 24841815, 24843415, 24843415, 24843315, 24841715, 24843915, 24842215, 24844415, 24844515, 24970315, 24970815, 24971015, 24845215, 24844516, 24842916, 24843516, 24840116, 24845215). All surgeries were performed using isofluorane anesthesia and following standards for aseptic surgery. Every effort was made to minimize suffering.

# Additional files

## Supplementary files

• Supplementary file 1. Real time PCR oligonucleotide sequences.

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
