## [Decision Letter]

Thank you for submitting your article "mTORC1 in AGRP neurons integrates exteroceptive and interoceptive food cues in the modulation of energy expenditure" for consideration by *eLife*. Your article has been favorably evaluated by Mark McCarthy (Senior Editor) and three reviewers, one of whom is a member of our Board of Reviewing Editors. The reviewers have opted to remain anonymous. The reviewers have discussed the reviews with one another and the Reviewing Editor has drafted this decision to help you prepare a revised submission.

Summary:

Burke and colleagues describe results from a series of studies investigating the effects of chemo-genetic manipulation of hypothalamic AgRP neurons on thermoregulation and thermogenesis. As previously reported, they found that this intervention in fasted mice results in reduced sympathetic nerve output to BAT as well as in reduced BAT temperature. Similarly, ghrelin injections yielded the same response in the absence of reduced thermogenic gene expression. Moreover, the authors demonstrate that both feeding and sensory food perception suppress the ability of CNO to reduce iBAT temperature. Furthermore, they demonstrate that refeeding, HFD-feeding and sensory food perception activated mTOR-phosphorylation in AgRP-neurons. Finally, they demonstrate that inhibiting mTORC1 regulates the AgRP-iBAT circuitry consistent with their proposed model. Intriguingly, the authors found that mTORC1 signal in AgRP neurons responds to food-related cues like caged food. Overall the paper is well written, and the experiments are well designed. Collectively, the results are well described and provide novel data. Nonetheless, several issues need to be addressed.

Essential revisions:

It is unclear whether AgRP neurons are involved in thermoregulation or iBAT thermogenesis. The fact that BAT temperature falls from ~36°C to ~34°C following CNO is somewhat surprising considering that BAT is specialized in producing heat. Could BAT temperature really become colder than the body set temperature of 37°C? Is the measured temperature really relative to BAT if it shows a systemic temperature falls following CNO, that could be related to torpor or other mechanisms? Did the authors monitor central/rectal temperature throughout the procedure? This is an important question to address, as thermoregulation and BAT thermogenesis are two independent (although overlapping) processes.

The mechanism by which mTORC1 signaling mediates AgRP effects needs further clarification. The authors claim that the lack of effect of increased mTORC1 signaling in AgRP neurons under chow maintenance reveals that the contribution of this pathway to energy expenditure is contingent on energy surfeit. However, one potential consideration is that the sustained activation of mTORC1 with HF diet results in several feedback inhibitions of insulin signaling that promote insulin resistance (by phosphorylating IRS in serine by example). One interpretation would be that inhibiting mTORC1 in a context of nutrient overload will improve insulin signaling, and thus is more likely to result in a phenotype compared to a situation where mTORC1 is working normally.

Could the authors comment on the fact that switching mice from chow to HF diet (31.8%kcal from fat) increases RER (Figure 4 and Figure 5—figure supplement 1)? How could HF diet increased RER to over 1.0 as it would be expected that to drop around 0.7?

The caveats of injecting 100% DMSO in the parenchyma of the brain should be discussed (e.g. osmotic stress, dehydration, toxicity).

In which AgRP population is mTORC1 active? Authors state that mTOR is activated in 15% of NPY/AgRP neurons. There should be an additional anatomical description and ideally co-labeling to indicate what this subpopulation is likely to be.

Validation of antibodies. mTORC1 activity was assessed in immunofluorescence with p-S2448-mTOR. Did the authors validate the specificity of this antibody? Phospho-S6 and phospho-S6K1 are a better readout of mTORC1 activity, although their validation would also be required. In addition, it is not p-mTORC1 that was evaluated but p-S2448-mTOR (subsection “mTORC1 in AGRP neurons mediates the thermogenic effect of leptin”, second paragraph).

The authors should appropriately put their results in the context of the published literature. When discussing the effect of AgRP-neuron ablation, both manuscripts simultaneously published (Luquet et al., as well as Gropp et al.) should be cited, similarly chemogenetic activation has been previously demonstrated to suppress BAT SNA in the absence of changes in thermogenic gene expression (Steculorum et al., 2016).

The authors demonstrate that feeding and sensory food perception abrogate the ability of AgRP-neuron activation to reduce BAT temperature. Under these conditions, is CNO still able to activate AgRP-neurons?

In Figure 6, the authors should show a monitor of BAT temperature following leptin administration, instead of subcutaneous body temperature. This would clarify the contributions of BAT thermogenesis versus other organs.

It is unclear why DN-S6K1 expression did not affect BAT thermogenesis and energy expenditure under a regular chow diet. Did the authors test if DN-S6K1 expression abolishes the effect of leucine administration on BAT thermogenesis?

[Editors' note: further revisions were requested prior to acceptance, as described below.]

Thank you for resubmitting your work entitled "mTORC1 in AGRP neurons integrates exteroceptive and interoceptive food-related cues in the modulation of adaptive energy expenditure in mice" for further consideration at *eLife*. Your revised article has been favorably evaluated by Mark McCarthy (Senior Editor), a Reviewing Editor, and two reviewers.

After discussion between the reviewing and senior editors, and the reviewers, the consensus is that the manuscript has been improved. However, there are some remaining issues that need to be addressed before acceptance, as outlined below:

The relative delay in the fall in the RER remains a concern. Several groups have published that the RER measures are lowered very quickly following exposure to high fat diet (often the same day). In addition, the RER values of over 1.0 (1.1) needs explanation. At a minimum, more experimental details are needed regarding the system used to assess RER and oxygen consumption.

In addition, the reviewers thought that not including studies monitoring core body temperature and BAT temperature was not adequately justified. The reviewers viewed the core body temperature measurements as important to delineate differences in thermogenesis vs. non-shivering thermogenesis.

---

## [Author Response]

*Essential revisions:*

*It is unclear whether AgRP neurons are involved in thermoregulation or iBAT thermogenesis. The fact that BAT temperature falls from ~36°C to ~34°C following CNO is somewhat surprising considering that BAT is specialized in producing heat. Could BAT temperature really become colder than the body set temperature of 37°C? Is the measured temperature really relative to BAT if it shows a systemic temperature falls following CNO, that could be related to torpor or other mechanisms? Did the authors monitor central/rectal temperature throughout the procedure? This is an important question to address, as thermoregulation and BAT thermogenesis are two independent (although overlapping) processes.*

We agree with the reviewers that thermoregulation and iBAT thermogenesis are overlapping processes, and understanding whether AGRP neurons regulate one or the other is important. We did not monitor core temperature simultaneously, as we did not have access to the equipment to simultaneously and remotely measure iBAT and core temperature (avoiding stress-induced responses that could be triggered otherwise). However, we think the evidence presented (in particular metabolic rate, and iBAT norepinephrine turnover) strongly indicate that CNO rapidly affects iBAT thermogenic activity, thereby supporting a role for iBAT thermogenesis in the response to AGRP activation. In addition, CNO-induced hypothermia was dramatically blunted at thermoneutrality, a condition that suppresses thermogenesis, further supporting a role for thermogenesis and iBAT in AGRP control of temperature.

Mice have a low thermal load and high temperature conductance compared to larger animals, causing their core temperature to reach ambient temperature within 20 minutes. Therefore, we think a fall of 2°C in this time frame is plausible.

iBAT is indeed specialized in producing heat, but can be turned off very rapidly, within minutes when animals are switched from 22°C to 30°C ^1^. The probe we used was located under the iBAT depot, within the subcutaneous compartment. If iBAT is switched off, the temperature of the probe may well reflect subcutaneous temperature and would not necessarily reflect core temperature. Even if the probe was reflecting core temperature, mouse core body temperature is highly plastic compared to humans, and the concept of core temperature set-point in small mammals reflects a range a temperatures between 35.5 and 38.5°C in mice ^2^.

We disagree that the response we are observing could be related to torpor, as torpor is characterized by immobility (not observed here) and body temperature lower than 30°C ^3^. In any case, suppression of iBAT thermogenic activity contributes to the metabolic adaptations necessary for torpor in mice ^4^. The other realistic mechanism that could contribute to the decrease in temperature induced by CNO is vasodilatation, although vasodilatation alone would not result in reduced energy expenditure. Here we observe a massive drop in energy expenditure following CNO administration, and under these conditions, suppression of iBAT thermogenic activity is the most likely underlying mechanism.

*The mechanism by which mTORC1 signaling mediates AgRP effects needs further clarification. The authors claim that the lack of effect of increased mTORC1 signaling in AgRP neurons under chow maintenance reveals that the contribution of this pathway to energy expenditure is contingent on energy surfeit. However, one potential consideration is that the sustained activation of mTORC1 with HF diet results in several feedback inhibitions of insulin signaling that promote insulin resistance (by phosphorylating IRS in serine by example). One interpretation would be that inhibiting mTORC1 in a context of nutrient overload will improve insulin signaling, and thus is more likely to result in a phenotype compared to a situation where mTORC1 is working normally.*

We agree that sustained activation of mTORC1 with HF feeding may result in several feedback inhibitions of insulin signaling that promote insulin resistance. In fact, overexpression of S6K1 signaling in the hypothalamus increases IRS serine phosphorylation ^5^. However, we disagree that this inhibitory feedback may contribute to our observations. Insulin hyperpolarizes AGRP neurons ^6^and therefore impaired insulin signaling would be expected to increase AGRP tone and suppress energy expenditure and iBAT thermogenic activity; in our model, overactivation of S6K1 produces an increase in these outcome measures. In addition, metabolic phenotyping of mice lacking insulin receptor in AGRP neurons reveals that insulin signaling in AGRP neurons does not modulate energy balance under chow or HF feeding ^7^.

Our data indicate that acutely, HF feeding increases mTORC1/S6K1 signaling in a subset of AGRP neurons, and that this increase contributes to but is not sufficient for the effect of a HF diet on energy expenditure. Thus, other events associated with HF feeding are necessary to allow the production of the metabolic consequences of S6K1 loss- and gain-of function in AGRP neurons.

*Could the authors comment on the fact that switching mice from chow to HF diet (31.8%kcal from fat) increases RER (Figure 4 and Figure 5—figure supplement 1)? How could HF diet increased RER to over 1.0 as it would be expected that to drop around 0.7?*

Figure 4 and Figure 5—figure supplement 1 are showing RER during the early phase following theswitch to HF diet. We agree these data indicate that the mice do not burn more fat during this early phase. We believe it is because the animals store fat massively instead of burning it, resulting in an average RER value similar to that measured when the animals where maintained on chow. We plotted average daily body weight gain against average RER over 24h before and after the switch and performed linear regression. Consistent with our interpretation, on chow the intercept of the regression is 1.02 and it drops to 0.73 after the switch.

Our data show that starting day 3 after the switch RER value begin to decrease (Figure 4), indicating that lower RER values would be reached once the animals fully adapt to the new diet.

*The caveats of injecting 100% DMSO in the parenchyma of the brain should be discussed (e.g. osmotic stress, dehydration, toxicity).*

We agree with the reviewers that 100% DMSO may produce side effects locally. We have not noticed any abnormal behavior or disruptions in locomotor activity, energy intake and energy expenditure following these injections. We have added discussion on the side-effects of DMSO parenchymal injections in the revised manuscript.

*In which AgRP population is mTORC1 active? Authors state that mTOR is activated in 15% of NPY/AgRP neurons. There should be an additional anatomical description and ideally co-labeling to indicate what this subpopulation is likely to be.*

We have added additional neuroanatomical information and co-labelling of the mTOR NPY-GFP co-labeling in the revised manuscript (Figure 3—figure supplement 1).

*Validation of antibodies. mTORC1 activity was assessed in immunofluorescence with p-S2448-mTOR. Did the authors validate the specificity of this antibody? Phospho-S6 and phospho-S6K1 are a better readout of mTORC1 activity, although their validation would also be required. In addition, it is not p-mTORC1 that was evaluated but p-S2448-mTOR (subsection “mTORC1 in AGRP neurons mediates the thermogenic effect of leptin”, second paragraph).*

We have included a validation experiment to confirm the specificity of the p-S2448-mTOR antibody that we used (Figure 3—figure supplement 1). We agree that phospho-S6K1 would be a better marker of mTORC1 activity, but we have been unsuccessful so far in using commercially available antibodies to detect this protein with an immunofluorescent stain. Phospho-S6 would not be an appropriate marker as it is downstream from multiple signaling pathways in neurons ^8^. Reference to mTORC1 activity have been corrected where appropriate.

*The authors should appropriately put their results in the context of the published literature. When discussing the effect of AgRP-neuron ablation, both manuscripts simultaneously published (Luquet et al., as well as Gropp et al.) should be cited, similarly chemogenetic activation has been previously demonstrated to suppress BAT SNA in the absence of changes in thermogenic gene expression (Steculorum et al., 2016).*

These references have been added to the revised manuscript.

*The authors demonstrate that feeding and sensory food perception abrogate the ability of AgRP-neuron activation to reduce BAT temperature. Under these conditions, is CNO still able to activate AgRP-neurons?*

We did not determine whether presentation of food or food-related sensory cues abrogates CNO-induced AGRP neuronal activation. We do not think measuring c-fos expression in these conditions would be very informative. In neurons expressing hM3dq, CNO produces massive and sustained neuronal activation, and increases expression of the immediate early gene c-fos starting approximately 60 min after the injection. In contrast, food-related sensory cues rapidly and transiently inhibit AGRP neurons ^9^. Because of the low temporal resolution of c-fos immunostaining as a marker of neuronal activation, and the difference in magnitude between CNO-induced activation and food-induced inhibition, we think it is unlikely that the transient inhibition induced by food presentation will be measurable with a c-fos stain. Following food ingestion, interoceptive signals produce a more sustained inhibition of AGRP neurons. However, even in this context, the supraphysiological nature of CNO-induced neuronal activation may mask food-induced inhibition at the c-fos level. Consistent with this idea, optogenetic activation of AGRP neurons increases c-fos immunostaining in AGRP neurons even when mice are allowed to eat ^10^.

*In Figure 6, the authors should show a monitor of BAT temperature following leptin administration, instead of subcutaneous body temperature. This would clarify the contributions of BAT thermogenesis versus other organs.*

We did not monitor iBAT temperature following leptin administration. We agree that multiple mechanisms may contribute to leptin-induced increase in body temperature, including the regulation of body temperature set point or thermal conductance, as recently proposed ^11,12^, but the temperature detected by a probe located under the interscapular brown fat pad may reflect changes in these processes as well. Thus, iBAT temperature measurements would not necessary rule out the contribution of peripheral thermoregulatory processes. Discussion on these limitations has been added to the revised manuscript.

*It is unclear why DN-S6K1 expression did not affect BAT thermogenesis and energy expenditure under a regular chow diet. Did the authors test if DN-S6K1 expression abolishes the effect of leucine administration on BAT thermogenesis?*

Lack of effects of DN-S6K1 expression on BAT thermogenesis and energy expenditure under a regular chow diet may be explained by the lack of complete inhibition of the S6K1 pathway using this strategy, as previously reported ^5^. Accordingly, residual S6K1 activity may be sufficient to maintain appropriate metabolic sensing under chow. We did not measure the ability of central leucine to increase iBAT thermogenesis under these conditions, but based on this, we predict L-leucine-induced increase in iBAT temperature would not be abolished. In contrast, the dominant negative mutant may sufficiently counteract the chronic increase in S6K1 activity under HF maintenance, hence producing a phenotype.

1) Cannon, B. & Nedergaard, J. Nonshivering thermogenesis and its adequate measurement in metabolic studies. The Journal of experimental biology 214, 242-253, doi:10.1242/jeb.050989 (2011).

2) Sunagawa, G. A. & Takahashi, M. Hypometabolism during Daily Torpor in Mice is Dominated by Reduction in the Sensitivity of the Thermoregulatory System. Scientific reports 6, 37011, doi:10.1038/srep37011 (2016).

3) Gavrilova, O. et al. Torpor in mice is induced by both leptin-dependent and -independent mechanisms. Proceedings of the National Academy of Sciences of the United States of America 96, 14623-14628 (1999).

4) Jastroch, M. et al. Seasonal Control of Mammalian Energy Balance: Recent Advances in the Understanding of Daily Torpor and Hibernation. Journal of neuroendocrinology 28, doi:10.1111/jne.12437 (2016).

5) Ono, H. et al. Activation of hypothalamic S6 kinase mediates diet-induced hepatic insulin resistance in rats. The Journal of clinical investigation 118, 2959-2968 (2008).

6) Qiu, J. et al. Insulin excites anorexigenic proopiomelanocortin neurons via activation of canonical transient receptor potential channels. Cell metabolism 19, 682-693, doi:10.1016/j.cmet.2014.03.004 (2014).

7) Konner, A. C. et al. Insulin action in AgRP-expressing neurons is required for suppression of hepatic glucose production. Cell Metab. 5, 438-449. (2007).

8) Knight, Z. A. et al. Molecular profiling of activated neurons by phosphorylated ribosome capture. Cell 151, 1126-1137, doi:10.1016/j.cell.2012.10.039 (2012).

9) Chen, Y., Lin, Y. C., Kuo, T. W. & Knight, Z. A. Sensory detection of food rapidly modulates arcuate feeding circuits. Cell 160, 829-841, doi:10.1016/j.cell.2015.01.033 (2015).

10) Chen, Y., Lin, Y. C., Zimmerman, C. A., Essner, R. A. & Knight, Z. A. Hunger neurons drive feeding through a sustained, positive reinforcement signal. eLife 5, doi:10.7554/eLife.18640 (2016).

11) Fischer, A. W. et al. Leptin Raises Defended Body Temperature without Activating Thermogenesis. Cell reports 14, 1621-1631, doi:10.1016/j.celrep.2016.01.041 (2016).

12) Kaiyala, K. J., Ogimoto, K., Nelson, J. T., Muta, K. & Morton, G. J. Physiological role for leptin in the control of thermal conductance. Molecular metabolism 5, 892-902, doi:10.1016/j.molmet.2016.07.005 (2016).

[Editors' note: further revisions were requested prior to acceptance, as described below.]

*After discussion between the reviewing and senior editors, and the reviewers, the consensus is that the manuscript has been improved. However, there are some remaining issues that need to be addressed before acceptance, as outlined below:*

*The relative delay in the fall in the RER remains a concern. Several groups have published that the RER measures are lowered very quickly following exposure to high fat diet (often the same day).*

In the dataset presented in this manuscript, RER values decreased starting 48h following the transition from standard chow to a diet containing 31.8% kcal as fat.

We agree with the reviewer that in some cases, RER values decrease within the first 24h following the transition from a low fat to a high fat diet. However, this is not always the case, as several factors affect changes in substrate utilization during this transition, including the percentage of fat in the diet, or whether the animals are on positive energy balance.

Here we used a diet with a relatively low fat content compared to standard high fat diets containing 60% of fat. In addition, as detailed in our first response, mice were at energy balance before the switch, and in positive energy balance after the switch, gaining on average 1.2g per day. Collectively our data indicate that during the first 48h following the switch, macronutrient-specific oxidation is not altered and the extra calories from fat are stored and not burnt.

Others have reported that increased lipid oxidation following the introduction of a high fat diet can be delayed to up to day 4 after the switch (Jackman MR et al., AJP 2010).

*In addition, the RER values of over 1.0 (1.1) needs explanation. At a minimum, more experimental details are needed regarding the system used to assess RER and oxygen consumption.*

RERs above 1 do occur physiologically as the metabolic process of de novo lipogenesis (converting glucose to FFAs) has an RER of 5.55 (Virtue and Vidal-Puig, Frontiers in Physiology, 2013). de novo lipogenesis can represent a sufficiently large proportion of metabolism to drive RERs above 1 (Arch JR et al., Int J Obes, 2006), particularly during the dark-phase for mice. While, we do agree the average RER values we have detected appear slightly high, we do not think this affects the differences observed between groups, as all groups were run in parallel using the same system and under the same conditions. All data were collected using a custom-built calorimetry system, described below:

The metabolic chambering system we have is an indirect calorimeter that measures both CO2 and O2 concentrations.

The metabolic chambering system has two banks of 4 gas analyzers (4 plex design). The system runs on an approximately 10 minute 40 second cycle (96 second presampling time to clear residual gas in the lines and a 36 second measurement time) with each chamber and a reference room air sample analyzed sequentially. The system uses individual mass flow controllers (Aalborg DFC26S-VADN5-C5A) to control the flow of air to each chamber. The air flow to each chamber is 400 ml/min at international standard temperature and pressure (~444 at US standard temperature and pressure). The chamber volume is approximately 12 litres. Chambers are routinely checked at the start and end of each run to confirm accurate flow rates and that they are operating continually under positive pressure. Flow into the gas analyzers is driven by a sample pump and then a separate mass flow controller set at 120 ml/min. The air is dried using Nafion drying tubes (Perma Pure) that run with a counter-current flow of dry air generated by a compressor. The gas analyzers are ordered in series and comprise of a paramagnetic O2 analyser (Servomex Pm1158) and an infrared CO2 sensor (Servomex IR1520-707).

Analyzers are calibrated with span gas (1% CO_2_) and room air for the upper calibrations and nitrogen as the zero gas. System performance is tested by infusing a 20% CO_2_/80% nitrogen test gas at 10 ml/min which should return an energy expenditure of 42 J/min at an RER of 0.94.

Energy expenditure is calculated from VO2 and VCO2 using a modified Weir equation. RER was calculated as VCO2/VO2.

*In addition, the reviewers thought that not including studies monitoring core body temperature and BAT temperature was not adequately justified. The reviewers viewed the core body temperature measurements as important to delineate differences in thermogenesis vs. non-shivering thermogenesis.*

To address this comment we have performed a study in which mice were implanted with a probe under the iBAT and a probe inside the peritoneal cavity to monitor iBAT and core temperature simultaneously. These data (added to the revised manuscript – Figure 1—figure supplement 1) show that iBAT temperature is an average 1°C higher than core temperature and drops first in response to AGRP activation. 15min after CNO injection, iBAT temperature is significantly lower than at baseline, while core temperature has not dropped yet. These data indicate that iBAT thermogenesis is rapidly turned off following AGRP chemogenetic activation, before changes in core temperature can be detected, which is consistent with published data showing that chemogenetic activation of AGRP neurons turns off sympathetic tone to iBAT within seconds (Steculorum et al., Cell, 2016).